# Chitin and Chitosan as Polymers of the Future—Obtaining, Modification, Life Cycle Assessment and Main Directions of Application

**DOI:** 10.3390/polym15040793

**Published:** 2023-02-04

**Authors:** Klaudia Piekarska, Monika Sikora, Monika Owczarek, Jagoda Jóźwik-Pruska, Maria Wiśniewska-Wrona

**Affiliations:** Lukasiewicz Research Network-Lodz Institute of Technology, Marii Curie-Sklodowskiej 19/27, 90-570 Lodz, Poland

**Keywords:** chitin derivatives, chitosan, LCA, biomimetic materials, medical application

## Abstract

Natural polymers are very widespread in the world, which is why it is so important to know about the possibilities of their use. Chitin is the second most abundant reproducible natural polymer in nature; however, it is insoluble in water and basic solvents. Chitin is an unused waste of the food industry, for which there are possibilities of secondary management. The research led to obtaining a soluble, environmentally friendly form of chitin, which has found potential applications in the many fields, e.g., medicine, cosmetics, food and textile industries, agriculture, etc. The deacetylated form of chitin, which is chitosan, has a number of beneficial properties and wide possibilities of modification. Modification possibilities mean that we can obtain chitosan with the desired functional properties, facilitating, for example, the processing of this polymer and expanding the possibilities of its application, also as biomimetic materials. The review contains a rich description of the possibilities of modifying chitin and chitosan and the main directions of their application, and life cycle assessment (LCA)—from the source of the polymer through production materials to various applications with the reduction of waste.

## 1. Introduction

### 1.1. Obtaining Chitin and Chitosan

Chitin (from Greek Chiton—tunic, covering) was first isolated from fungi in the form of an alkali-resistant fraction by Branconnot in 1811 [1,2,3,4]. It is a biopolymer produced by natural biosynthesis and is the second (after cellulose) most common reproducible natural polymer in nature. Chitin is one of the animal polysaccharides, as it is the main building component of insect skeletons and the shells of crustaceans and arachnids [5,6]. It is also found in the cell membranes of bacteria, molds and fungi, i.e., yeast [7,8], and in sponges [9] and corals [10]. Chemically, the chitin molecule consists of 2-acetylamino-2-deoxy-D-glucose units linked by β-glycosidic bonds in the 1,4 position. In nature, it occurs in the structural elements of the exoskeleton as an ordered structure of crystalline microfibrils; it also has three polymorphic varieties: α, β and γ [11]. The mentioned forms of chitin differ in the way that the chains are packed in the crystalline areas, the degree of hydration, the size of the elementary cells and the amount and degree of polymerization of the polymer chains [12,13]. The most stable and widespread in nature is the α form, contained, for example, in the shells of crustaceans, skeletons of insects or mushrooms. The β form, which is much less common, is present, for example, in squids [14]. In the α form of chitin, there is an antiparallel arrangement of molecules in the chitin chain and strong hydrogen bonds between the chains are formed, stabilizing the polymorphic structure. The β form of chitin is characterized by parallel packing of molecules and the presence of weaker hydrogen bonds, while in γ chitin, alternately with one antiparallel molecule, there are two parallel molecules [15]. Differences in the crystal structure of both amorphous forms of chitin directly affect their further processing capabilities. The ordered structure and strong intra- and intermolecular interactions of chitin limit its solubility, and thus reduce the possibility of its use in industry. Due to the spatial arrangement of the 2,3-trans substituents in the monosaccharide unit, chitin is very stable for most reagents, and regardless of the polymorphic form in which it occurs, it does not dissolve in water or common solvents [1], but is degraded under the influence of the chitinase enzyme, produced by some microorganisms. Studies aimed at finding appropriate solvent systems for chitin indicate the effectiveness of ionic liquids [16], calcium chloride–methanol system [17], ice-cold alkaline mixture [18], N,N-dimethylacetamide (DMAc)/lithium chloride (LiCl), 5% (*w*/*w*) [19] and some strong acids and fluorinated solvents [20]. However, they could not be implemented in the industry; all the mentioned solvent systems had various contraindications, i.e., corrosive, toxic or volatile properties, and often caused a strong degradation of chitin. A method of dissolving chitin was developed, consisting of introducing it to a mixture of 8 wt. sodium hydroxide/4% wt. an aqueous solution of urea and cooling the mixture to −20 °C [21,22]. The alkalichitin obtained in this way, i.e., a dissolved form of chitin, is an environmentally friendly material, and has found potential application in the field of biomaterials and biomedicine [4]. About 10^12^ tons of biosynthesized chitin are used annually [6]. However, for research and industrial works, it is obtained mainly from marine invertebrates, i.e., shrimps, crabs, lobsters or krill, or directly from the walls of fungi. Regardless of the source, chitin is an unused waste of the food industry. Scientists are largely focused on the secondary use of this raw material. The process of obtaining chitin is quite time-consuming. The chitin deacetylation reaction is shown in Figure 1.

The first stages are the cleaning and fragmentation of crustacean shells, then the extraction and demineralization stage—treatment with concentrated hydrochloric acid in order to deprive them of calcium carbonate and calcium phosphate—and the stage of deproteinization (removal of proteins), i.e., another treatment with liquid sodium hydroxide, and the final stage of elimination of carotenoid pigments, i.e., discoloration. The specific conditions for the chitin isolation process depend strictly on the source of its origin [23]. To sum up, the limited solubility and complicated processing of chitin mean that it is used to a small extent and only in strictly defined areas.

As a result of the process of alkaline deacetylation of chitin, or alternatively, enzymatic deacetylation, one of its most valuable derivatives is obtained—chitosan [24]. The source of chitin, its properties and the conditions of the deacetylation process (temperature, time and concentration of alkalis) have a significant impact on the chemical purity and quality of parameters related to the biological activity and technological usefulness of chitosan [25]. The properties of the obtained chitosan are mainly described by the degree of deacetylation and the molecular weight. The degree of deacetylation (DD) is one of the most important chemical parameters, thanks to which we can distinguish chitosan from chitin; it is defined by the ratio of the number of -NH_2_ groups formed to the initial number of -NH-CO-CH_3_ groups present in chitin, i.e., is a statistical interpretation of the product obtained after the deacetylation process and characterizes the arrangement of chains in the biopolymer macromolecule. The DD value determines the properties of this natural polymer, such as its solubility in aqueous acidic solutions, the degree of swelling in water, susceptibility to biodegradation, biological activity and biocompatibility [4]. For the alkaline deacetylation of chitin, a 50% lye is used, usually sodium or potassium hydroxide solution, where it has to stay for several hours at a temperature of 100 °C or above, while during the enzymatic treatment, chitin is treated with a hydrolytic enzyme—N-deacetylase. The degree of deacetylation of commercially available chitosans ranges from 60% to 100% [26]; for comparison, chitosan isolated from filamentous fungi belonging to the order Mucorales is characterized by a degree of deacetylation at the level of 30 to 47% [27,28]. A higher degree of deacetylation means better quality. From a chemical point of view, chitosan is deacetylated chitin, and the chitin deacetylation process consists in alkaline hydrolysis of amide groups in the poly[-(1,4)-2-acetamido-2-deoxy-D-glucopyranose] chain and partial removal of acetyl groups from acetylamino groups in the chitin chain and replacing them with amino groups. Multiple deacetylations lead to obtaining chitosan with a higher degree of deacetylation. The average molecular weight (Mw) of native chitin exceeds one million, while commercially available chitosan ranges from 100,000 to 1,200,000. Many literature reports say that prolonging the deacetylation process increases DD while reducing the average molecular weight [4]. There is also a dependence of crystallinity on the degree of deacetylation. Chitosan is a polymer with a low crystallinity, which decreases even more with the increasing degree of deacetylation of chitosan, both in the method of heterogeneous and homogeneous deacetylation. In the case of heterogeneous deacetylation, the sample with DD = 89% is already completely amorphous. For homogeneous deacetylation, the decrease in the crystallinity of the samples is much more pronounced; samples with DD = 40% are already amorphous [29]. Due to the presence of free amino and hydroxyl groups, chitosan is more reactive, which means that it is susceptible to chemical modifications, which results in higher biological activity. Although chitosan is not soluble in water, but only in dilute aqueous solutions of organic acids, it has a hydrophilic character, which means that it absorbs water from the gas phase or when immersed in it. In order to prevent it from absorbing water, it can be subjected to modifications such as laminating, creating shells or mixing with other polymers [30]. However, in order to dissolve chitosan, it is necessary to lower the pH of the solution below 6.2 [31].

The factors determining the quality and functional properties of chitosan are the average molar mass, polymorphic structure, viscosity of its solutions and polymer purity [32]. Chitosan, like many other natural polymers, is characterized by polydispersity, i.e., heterogeneous distribution of molar masses (Mm), which is the result of depolymerization occurring at the stage of chitin deacetylation. As a consequence, the obtained chitosan is a mixture of biopolymer particles of various sizes. Most methods for determining the average molar mass are based on high-performance separation techniques, such as gel permeation chromatography. To determine this quantity in a solution, the least complicated and fastest method is the viscosimetric measurement [11]. When selecting the parameters for the determination of the average molar mass of chitosan, the intrinsic properties of the polymer and experimental factors should be taken into account. Experimental factors that have a significant impact on the assay are the pH and ionic strength of the test solution. The properties of the solvent affect its interaction with the diluted biopolymer, determining its conformation and hydrodynamic properties [33]. In dilute solutions of acids, chitosan macromolecules with different DD values can occur in the form of compact spheres, for which the exponent α of the Mark–Houwink equation is equal to zero. When α is in the range of 0.5 ÷ 0.8, chitosan macromolecules take the form of random coils, and at α 1.8, polymer macromolecules form rods. It has been proven that chitosan macromolecules with a molecular weight of less than 220 kDa form a conformation of random coils (α > 0.65), and those with a molecular weight of more than 220 kDa have a more compact shape (α < 0.6). This phenomenon is attributed to the fact that the high-molecular-weight polymer has more intramolecular hydrogen bonds or a more uniform charge distribution compared to the low-molecular-weight polymer [32].

Chitosan derivatives are created by various chemical reactions described below; it is worth mentioning that they are widely used in many areas of our lives.

### 1.2. Methods of Chemical Modification of Chitosan

The chemical modification of a polymer generally consists of changing the properties through a controlled change in the chemical composition of macromolecules. A change in the composition may occur during the introduction of new functional groups, their transformation, intramolecular cyclization or by reaction, i.e., oxidation, reduction, grafting, cross-linking or degradation. Modification under the influence of chemical factors can take place directly during the polymer synthesis or can be made on the finished polymer [34,35,36]. The molecular structure of chitosan, thanks to active functional groups, can be modified to form derivatives with improved physical, chemical and/or physiological functions.

The diagram (Figure 2) shows grouped types of chemical reactions that lead to the formation of chitosan derivatives. In the group of substitution reactions, i.e., the introduction of side groups to the chitosan chain according to various types of reaction mechanisms, there are reactions such as alkylation, acylation, arylation, thiolation, sulfonation, or phosphorylation, etc. [36,37,38,39,40,41,42,43,44,45,46,47,48]. The most frequently studied chitosan derivatives are carboxymethyl chitosan [45,49]. Carboxymethyl chitosan belongs to the group of amphoteric polyelectrolytes, as it contains two types of functional groups in the chain: basic amino groups and acid carboxylic groups [45]. The carboxymethyl group can replace both of these functional groups or one of them (amino or hydroxyl). The product of the carboxymethylation reaction will be an O-, N- or O,N-substituted derivative. For example, the reaction of monochloroacetic acid in an isopropanol solution in a strongly alkaline environment at 55 °C leads to the formation of O-carboxymethyl chitosan. In turn, N-carboxymethyl chitosan is obtained in the reaction of glyoxylic acid in the presence of sodium borane at 60 °C and pH 3.2–4.0 [50]. Compared to chitosan, carboxymethyl chitosan belonging to chitosan ethers is characterized by better solubility in water and better physicochemical and biological properties, i.e., antibacterial properties, biocompatibility or lack of toxicity.

In the acylation reaction, amino groups and/or hydroxyl groups can also be substituted. Depending on the reaction conditions and the nature of the modifying agent, the product may analogously be N-, O- or N,O-acyl derivatives of chitosan. Niemczyk et al. refer to the characterization of acyl derivatives using fatty acids [51,52]. Derivatives and their properties were characterized for use in biomedicine using various tests, i.e., tests of hydrolytic and enzymatic stability, tests of microbiological degradation or antimicrobial activity. Studies of susceptibility to hydrolytic degradation gave information that the degree and place of chitosan substitution affect the speed and course of this process. Another substitution reaction is chitosan thiolation. As a result of this reaction, sulfate derivatives of chitosan are formed, i.e., chitosan-cysteine, chitosan-thioglycol. The -SO_3_H functional group can attach to the -NH_2_ group in the glucosamine molecule to form N-substituted chitosan or to the -OH and/or -CH_2_OH group to form O-substituted chitosan. The type of reaction depends on the use of factors, i.e., reaction medium, concentrated sulfuric acid, sulfur dioxide and trioxide or chlorosulfonic acid [48,49,50,53,54,55]. Shastri [56] reported several applications of thiolated chitosan as nanocarriers for ophthalmic drug delivery. The modification also made it possible to obtain anticancer drugs with more effective mucoadhesion and greater ability to penetrate the membrane, used in nanomedicine [57]. Thiolated chitosan derivatives are also used in environmental protection by improving the efficiency of removal of Cu^2+^ and Cd^2+^ ions from municipal and industrial wastewater [58,59].

Chitosan as cationic natural polymer forms chelate complexes with metal ions. The complexing abilities of chitosan require the involvement of -OH and -O- groups in D-glucosamine residues as ligands, or otherwise two or more amino groups from a single chain binding the same metal ion. The free amino groups in chitosan are considered to be much more effective for metal ion complexation than the acetyl groups in chitin. Despite this, an increase in the content of free amino groups does not directly increase the derivatization capacity due to the influence of other properties, such as crystallinity, affinity for water and/or decomposition of the remaining units [60]. In chitosan phosphorylation reactions, the modifying agent may be a phosphorus-containing compound (e.g., orthophosphoric acid (V), orthophosphoric acid (III)). Negm et al. carried out a phosphorylation reaction using phosphorus pentoxide in methanesulfonic acid as a solvent, producing water-soluble products with a high degree of substitution [37]. The reaction of chitosan and orthophosphoric acid at 150 °C in the presence of urea as a catalyst in DMF also leads to the synthesis of phosphorylated chitosan [60]. Ramos et al. synthesized N-methylenephosphonic chitosan using chitosan, phosphorus and formaldehyde [61]. Combination with methylenephosphonic groups enhances the aqueous solubility under mild conditions of the obtained derivative. In the literature, we can find many methods of obtaining phosphated chitosan [53,54]. It should be mentioned that chitin is also subject to phosphorylation reaction, creating the so-called P-chitin, which is highly soluble in water, bioresorbable and biocompatible [62]. In combination with alginate, it is a good material for the production of coatings that can be used for controlled drug release, tissue engineering and other environmental applications. Both alginate and P-chitin are anionic polymers, thus allowing easy cross-linking with Ca^2+^ ions. Graft copolymerization and cross-linking belong to the group of reactions that extend the chain of the chitosan micromolecule, and at the same time lead to obtaining derivatives with a higher molecular weight. Graft copolymerization is one of the main methods of chemical modification of chitosan, consisting of grafting monomers (side chains), i.e., acrylic acid, methacrylic acid, acrylamide, acrylonitrile, styrene, aniline, lactic acid, etc., to the main chain of chitosan. The properties of graft copolymers depend to a large extent on the chemical nature of the side chains, their length (average molecular weight) and their number in a given chain (copolymer content). An example here is chitosan/lactic acid graft copolymer in the form of nanoparticles, which can serve as a carrier of drugs with prolonged action. In vitro studies have shown that the introduction of a lactate moiety into the chitosan skeleton reduces the rapid release of the tested active substances compared to chitosan alone [63].

Chitosan as a polyelectrolyte has the ability to form gels with high biocompatibility and nontoxicity, as well as high sorption capacity. Studies on chitosan cross-linking concern the formation of a polyelectrolyte complex with the use of substances capable of creating interchain cross-linking bonds of the nature of covalent and/or ionic bonds. Sodium alginate may be such an agent [64]. Chitosan and alginate are characterized by high biocompatibility and biodegradability, which allows them to be used in medicine as scaffolds in tissue engineering, dressings or carriers of medicinal substances, as matrices for cell culture and for use in the cosmetics industry, agriculture and environmental protection [65]. The formation of the chitosan/alginate complex is possible due to the formation of ionic bonds between the functional groups of positively charged chitosan (polycation) and negatively charged alginate (polyanion). The reaction of the biopolymer with aspartic acid in the environment of ethylene glycol, which acts as a high-boiling solvent and a cross-linking agent, can also lead to the formation of a hydrogel [32]. Chitosan as a biosorbent can be formed in cross-linking reactions, in which the type of cross-linking agent and the degree of cross-linking affect the ability to remove heavy metal ions, such as mercury or lead. The cross-linking density is affected by the type of cross-linking substance and its concentration, the degree of deacetylation and the average molecular weight of chitosan, the concentration of the chitosan solution (in the case of cross-linking in solution) and the cross-linking time. Cross-linking agents such as dialdehydes (glutaraldehyde, glyoxal), genipin and epichlorohydrin are used for chemical cross-linking of chitosan. It is also possible to use two types of cross-linking agents, i.e., factors leading to both covalent and ionic cross-linking at the same time, enabling an increase in the mechanical strength of the polymer while maintaining its hydrophilic properties [25]. A very good chitosan cross-linking agent, especially when this polymer is intended for biomedical applications and due to its biocompatibility and very low cytotoxicity, is genipin [66,67,68,69,70,71]. Genipin is a natural compound obtained from Gardenia jasmonides, and the mechanism of the chitosan cross-linking process depends on the pH of the environment [70,71]. This compound is about 5000–10,000 times less toxic than glutaraldehyde [71].

The last discussed group of chitosan modification methods is depolymerization.

The depolymerization method may be influenced by chemical, physical or enzyme factors [44,72,73]. With regard to chitosan, it consists of the breakdown of the polymer into oligomers and monomers in the presence of various acids, such as nitric acid, hydrochloric acid, phosphoric acid and free radicals (e.g., from the decomposition of dihydrogen dioxide, potassium persulfate or formed in the ozonation reaction). As a result of depolymerization, low-molecular-weight chitosan and chitosan oligomers with varying degrees of N-acetylation are obtained, which are of interest to medicine, biotechnology, the cosmetics industry, agriculture and other fields. The degradation process is the result of the breakdown of glycosidic bonds in the chitosan chain. Chemical degradation processes of chitosan occur according to various mechanisms: acid hydrolysis, oxidation and reduction reactions and a specific mechanism involving nitric acid (III), and are nonspecific methods, as they lead to the production of various types of oligomers and a large amount of d-glucosamine monomer. A more specific method of chitosan degradation is enzymatic depolymerization, which takes place in the presence of various enzymes, e.g., papain, lipase, chitanase [73]. The chitosan depolymerization process also occurs under the influence of such factors as thermal energy, gamma radiation, ultrasound (ultrasonification), plasma or microwave radiation [44]. The latter method is an alternative to conventional heating, giving a higher yield of the product and significantly shortening the synthesis time [1]. The depolymerization reaction is often accompanied by other side reactions, leading to the formation of various types of derivatives, especially if the depolymerization process takes place according to the free radical mechanism.

## 2. Biomimetic Materials as Inspiration for Functional Materials with Chitosan

Bionics (other names biomimetics, biomimicry) is a dynamically developing discipline that studies biomaterial systems and their structure, functionality and optimization as a result of the use of various fields of science, i.e., biology, chemistry and physics. Active substances from natural renewable raw materials are used to design various structural and functional materials using the principles of bionics found in the natural environment [74,75,76]. In the vast majority of the analyzed cases, bionics does not involve direct copying of the structure of biological materials, only using the guiding principles of bionics from natural systems in order to artificially synthesize materials with desired properties and functionality [74,77]. This subsection of the article focuses on imitating the unique structure and functionality of some organisms found in nature, e.g., mussels, biological cell substrates, lotus flowers, desert beetles or honeycomb structures, when designing functional chitosan-based materials with similar properties (Figure 3). Biomimetic materials, taking into account their unique properties, i.e., adhesion, slow release and adsorption, are used, among others, in in biomedicine, tissue engineering, adsorption materials and other fields [74].

Marine mussels are characterized by a stable structure that allows them to adhere to various materials in the aquatic environment. This is mediated by the mussel foot protein sequence (MFPS) containing a large amount of 3,4-dihydroxyphenyl-L-alanine (DOPA) [78]. The catechol group plays the key role in the mechanism of adhesion. A mixture of chitosan and catechol is an excellent raw material for the production of functional adhesive materials, used, among others, as a material for bioprinting, e.g., of bone inserts in tissue engineering, as a drug nanocarrier in targeted oncology therapies, or for functionalization of the surface of new biomaterials [79,80,81]. Biomineralization through the interaction of a specific surface bond at the organic–inorganic interface refers to a biological process that results in the formation of highly ordered biological materials, i.e., shells, pearls, enamel or bones. Biominerals are characterized by a hierarchical layered structure, as well as specific mechanical and physical properties, which allows for functional diversification of the obtained biomaterials. The mature mother of pearl consists of very thin matrix layers of about 30 nm and thicker layers of calcium carbonate and aragonite—lamellae (about 500 nm). In recent years, scientists have used a number of innovative techniques to simulate the microstructure of the nacre layer and produce materials with good mechanical properties [82,83]. Based on the structure of nacre, Almeida et al. produced multifunctional films containing chitosan and hyaluronic acid, developed in dip-coating technology. Inorganic and bioactive glass nanoparticles were alternately applied with layers of natural polymers: chitosan, hyaluronan, catechol. The films obtained in this way are used in bone tissue engineering due to the possibility of obtaining environmental conditions compatible with osteogenesis [83]. Other researchers used chitosan as a matrix and alumina layers as reinforcing particles to prepare a film material with a pearlescent microstructure. The same researchers confirmed the influence of the proportion of organic and inorganic matter as well as relative humidity on the mechanical properties. Studies have shown that the relative humidity of the environment had a key impact on the measured mechanical properties of the tested films [84]. Yao et al. developed an innovative solution for obtaining biomimetic, hybrid composite membranes containing chitosan and montmorillonite (MMT) obtained by self-assembly as a result of vacuum filtration or water evaporation. The resulting blocks were arranged in a pearl composite material with a clear layered structure. The obtained film had high mechanical strength and was characterized by light transmission and fire resistance [85,86]. Xie et al., using carboxymethylated chitosan and MMT, obtained a transparent and fire-resistant nanocoating with a nacreous microstructure as a result of a self-assembly process [87]. Fang et al. demonstrated a similar solution to obtain a mother-of-pearl flame-retardant system by the LbL method using MMT as a matrix and chitosan. The resulting composite in the form of paper had excellent flame-retardant properties [88]. Saito et al. obtained a solution that can be effectively used in monitoring during the treatment of burn wounds. They produced tetracycline-containing nanoparticles characterized by high fluorescence and a high degree of transparency, which facilitated observation during burn treatment. The sheets contained chitosan and alginate applied by the LbL method, followed by an antibiotic layer and a bottom layer composed of polyvinyl acetate as a hydrophobic barrier layer [89]. The extracellular matrix (ECM) is responsible for directing cellular development. It is the structural basis of cells and is the source of three-dimensional biochemical and biophysical signals that trigger and regulate the behavior of the cell [90,91]. Hydrogel materials developed on the basis of chitosan can imitate the shape and function of natural ECM in vivo by adjusting their physicochemical and mechanical properties [92,93], which natively lead to cellular functions. The construction of functional tissues is based on the cell–biomaterial interaction. In their research, Maria et al. created a permeable polymer network (IPN) as a result of combining chitosan and hyaluronic acid polymers with a terpolymer containing the active substance catechol, cross-linked with Fe. It was found that this hybrid system can be used as a wound dressing due to its high affinity for cells, high bioactivity and above all, stimulation of the extracellular matrix. In addition, the controlled release of the active substance in the form of catechin promoted the process of tissue regeneration and contributed to wound healing [94]. Another example of the impact of chitosan on the extracellular matrix is the modification of the surface of chitosan fibers with fragments of human collagen I in a physical and chemical way to obtain a complex of glycosaminoglycans (GAG) and peptides similar to those present in the matrix cell [95]. The research showed that compared to the physical modification, the chemical modification resulted in an even distribution of the peptide on the fiber without changing its shape. Orthopedic biomaterials or coatings with ECM-like nanofunctions can induce proper interaction between the bone tissue and the implant surface. An example of such a promising composite material is chitosan–gelatin, because it combines the cell adhesion of gelatin with the antibacterial properties of chitosan [96]. However, Tangprasert et al. developed a gelatin/chitosan/complex calcium phosphate-based hydrogel that was used to simulate the extracellular matrix of calcified soft tissues. In the next step of their research, they designed an ex-vivo model to assess tissue formation. The results showed that the molecular structure and morphology of the self-assembled hydrogel was similar to the extracellular matrix formed by bone in situ, and its physical and biological properties enhanced cell viability and proliferation [97]. Many research teams are inspired by the Stenocara gracilipes beetle that lives in the African Namib Desert. Although the Namib is one of the hottest and driest places in the world, the beetle is able to capture moisture from windblown fog droplets. This is due to the presence of small bumps on the wing coverts of the beetle; these are nonwaxy, hydrophilic raised areas that allow for the collection of microscopic droplets, while the moisture-repellent hydrophobic areas below act as channels through which water flows to the mouth [98,99]. These features, together with the advantages of polyvinylidene fluoride and chitosan, were used in his research by Al-Gharabli [100], who achieved better results of separation of hybrid materials by adopting the “grafting” method. In order to fix the chitosan on the surface and internally, a silicon-alkyl modifier with a porous structure was used. It was found that the hydrophilic nature of chitosan can improve membrane permeability, increase antifouling properties and broaden the range of applications of new materials. Based on the naturally occurring honeycomb structure with the most closely filled hexagonal holes, Zhang et al. developed a new ceramic–polymer composite using chitosan and poly(ethylene). In contrast, the three-dimensional porous honeycomb carbon has excellent mechanical properties and a defined surface [101]. Dai et al. used this property to prepare a three-dimensional composite system composed of chitosan/honeycomb porous carbon/hydroxyapatite. The composite scaffolding produced had high porosity and specific mechanical strength similar to that of bone tissue [102]. It was found that as a result of the gelation and carbonization process, chitosan can form a composite material with a porous honeycomb structure, which showed good adsorption of toxic Cr(VI) metal ions and decomposed organic water impurities. In order to alleviate the increasingly serious problems of environmental pollution, Zou et al. reported a novel environmentally friendly material with high thermal conductivity and honeycomb structure, which was obtained using a significant difference between chitosan microspheres and hexagonal hydroxyl-functionalized boron nitride nanoparticles [103]. The new heat-conducting materials were degradable, rapidly recyclable and had broad market prospects.

## 3. Applications

### 3.1. Application of Chitosan in Cosmetic Industry

The definition of cosmetic describes it as any substrate or preparation designed for the contact with the various external parts of the body, teeth and the mucous membranes of the oral cavity [104]. Chitin, chitosan and their derivatives are among active ingredients widely used in the cosmetics industry. Their use in different body sites, including skin, hair, gums and teeth, is described in literature. Chitosan has found an application in all fields of dentistry [105]. Bioactivity, anti-inflammatory properties, hemostatic activities and wound healing are among the most important properties in this field.

The most common oral disease is dental caries, which still remains a major dental health problem [106]. Cariogenic bacteria, including both S. mutans and lactic acid bacterial species, play a crucial role in the pathogenesis of the disease. They are able to grow in acidic conditions resulting from the digestion of products such as sugars [107]. Because of antimicrobial activity, chitosan was tested as an alternative product used against oral pathogens, with fewer side effects than widely used chlorhexidine gluconate, metronidazole, ampicillin or quaternary ammonium compounds [104,108]. Aliasghari et al. compared the inhibitory effect of chitosan and chitosan nanoparticles against *S. mutans*, *S. sobrinus*, *S. sanguis* and *S. salivarius* [109]. The conducted study showed lower minimal inhibitory concentration values for chitosan nanoparticles than for chitosan. The smaller size of nanochitosan can be connected with the higher affinity for bacterial cells and higher antibacterial activity. Yet, it should be mentioned that both chitosan and nanochitosan showed antigrowth and antiadherence effects against bacteria, highlighting a great potential of the biomaterial in the prevention of dental caries.

The gel form of chitosan has found an application in the treatment of chronic periodontitis and canker sores [110]. Akıncıbay et al. [111] investigated the clinical effectiveness of chitosan (gel form and as active agent) in the treatment of chronic periodontitis. The conducted study used the bioactive biopolymer as the vehicle for local delivery of metronidazole. The authors concluded that chitosan seems to be a promising material in such application.

Chitosan-based modifications are also widely used in oral hygiene products, including toothpastes [112,113], mouthwashes [114,115,116] or varnish. A great part of cosmetic industry are hair care products. The literature reports about the application of chitosan derivatives in a large variety of such products, including shampoos, rinses, hair colorants, permanent wave agents, styling lotions, hair tonics and sprays [104,117]. The main advantage of chitosan and its derivatives that predisposes its use in hair care is the ability to interact with keratin, forming transparent and elastic films over hair fibers. It influences hair softness and strength and helps to avoid hair damage. The application of the triple-component blends composed of chitosan, hyaluronic acid and collagen resulted in the improvement of the mechanical properties, general appearance, and condition of hair [118]. The moisturizing effect and film former activity allowed for the use of various forms of chitosan as additives in shampoos and hair sprays [104].

Being antimicrobial, skin-protectant, emollient, cleansing, antioxidant, conditioning and humectant are among the chitosan functions that are desired in skin care products [119]. Special interest is focused on antiaging cosmetics. Aging is a natural process that involves the degradation of the extracellular matrix in the epidermal and dermal layers and is connected with changes in biochemical composition, including a reduction in collagen production, hyaluronic acid and water amount [120]. Chen et al. aimed to develop an effective cosmetic based on natural polymers acting against skin aging [121]. They applied carboxymethyl chitosan/organic montmorillonite nanocomposite as a component of cosmetic formulation to prepare cream. Study revealed good moisture absorption and retention compared to hyaluronic acid. Chitosan is also an ingredient in color cosmetic products, including lipstick, makeup, eye shadow and nail polish [119].

### 3.2. Application of Chitosan in Packaging and Food Industry

Chitosan has found an application in the food industry and is an approved ingredient in Korea, Japan, the USA and Europe [122]. It plays a role as a food preservative, acting as a natural antioxidant, improving food safety quality and shelf life. Its flocculating ability is used in the clarification of solutions, such as juice [119].

Due to antimicrobial activity, especially against wide range of foodborne filamentous fungi, bacteria and yeast, chitosan can be used as a food preservative [123,124,125]. The antibacterial activity principle of chitosan is still unknown and still requires deep analysis. The literature considers the hypothesis linking the activity with the leakage of proteinaceous and other intracellular constituents, which results in changes in the cell permeability barrier because of interactions between the positively charged chitosan molecules and negatively charged microbial cell membranes [124]. Another theory is connected with the ability of chitosan to absorb oxygen, which is crucial for filamentous fungi and acetic acid bacteria [126]. The application of chitosan as a preservative was described in milk, wine and fruit juices [127,128,129,130]. Yet, it should be mentioned that the type of matrix influences the efficiency of the agent. Some beverages are expected to be clarified products. The process requires the introduction of a clarification step and is connected with the removal of suspended solids with different origins. It mainly depends on the nature of the product and its properties such as charge density, the presence of functional groups, and molecular weight [131]. Protonated chitosan interacts with negative compounds through electrostatic interactions. The behavior allows for the use of the biopolymer as a clarifying agent of fruit juices, wines, beers and tea [131,132,133,134]. A study revealed that clarification with chitosan is more efficient with lower doses than bentonite and gum arabic [135].

The use of chitosan in encapsulation is also widely described in literature. The main advantage that determines the application of the biopolymer in the fabrication of coatings is the ability to form films. Yet, it should be mentioned that chitosan films have a great permeability to water/gas, which may restrict their use in the food industry [136]. The main aim of encapsulation is protection from oxidation and the improvement of the bioavailability of active compounds [137]. Encapsulation protects labile compounds and probiotics from the environmental conditions of the matrix and gastrointestinal environment. Additionally, it improves the water solubility of lipophilic compounds [131]. The literature reports about the various forms of chitosan that have been applied in the encapsulation of a diverse range of compounds, including nanoparticles, nanohydrogels, nanofibers and nanocomposites [136].

The designed product demonstrated a superior performance against Aspergillus niger. The study confirmed that the connection of chitosan and clove oil can be used as a natural fungicide in agricultural and food industry. Piran et al. [138] encapsulated green tea extract in chitosan nanoparticles and aimed to control its antioxidant activity. The study revealed the increase in the activity of green tea after coating with the biopolymer, making the composition promising in the delivery of green tea polyphenols. The encapsulation with the use of chitosan allows simultaneously for the enhancement of lipid solubility and antioxidant activity, maintaining the stability of compounds. Other scientists [139] described the encapsulation of polyphenols extracted from grape and apple pomace in chitosan and soy protein. The efficiency of the process was high, and reached 75% and 95%, respectively. Both coats showed a protective effect on polyphenols, which were kinetically released from them. The potential use of the composition as a food antioxidant material was confirmed.

In the last few decades, environmental issues have attracted increasing interest from scientists, consumers and thus, producers. Packaging, including food packaging, was reported to be one of the largest application fields for plastics [140,141]. Due to this fact, there is a great need to look for ecological alternative solutions for this sector. In the literature, there are many reports about the application of chitosan in the packaging of food. Chitosan-based composites can be used either as films or edible coatings, which are known to extend the shelf life of food products [142].

Coatings can be either directly applied onto the surface of food (edible) or onto the surface of packaging material (functionalization) [142,143]. The main aim of the use of coatings is to retard ripening and water loss and reduce the decay of fruits and vegetables. Additionally, in meat products, they can delay moisture loss, enhance product appearance and reduce lipid peroxidation and discoloration. Coatings can be prepared by dispersion of a liquid film onto a food product [142,144]. In the literature, three different types of coating have been described: spread coating (uses a sterile spreading tool such as a brush or spatula), spray coating (uses several spray tools such as compressed air-assisted sprayers) and dip coating (requires the immersing of the food material into chitosan-based solutions) [145]. The combination of chitosan with other polysaccharides (e.g., starch, alginate) results in the improvement of mechanical properties, lower water solubility and better performance in terms of water vapor permeability [142,146]. Examples of chitosan blends that were described in the literature in food packaging are presented in Table 1.

Chitosan-based composites deserve attention in the packaging industry. They are widely studied due to their advantages, such as biocompatibility, biodegradability, nontoxicity, and renewability. Yet, the improvement of their performance in terms of thermal, mechanical and water barrier properties is needed. Additionally, the high cost of production when compared to the traditional petroleum-based plastics influences its limited application [160,161].

### 3.3. Application of Chitosan in Medicine

Interest in the use of chitosan in the biomedical area is constantly growing, mainly due to its unique biological properties. Due to the ability to biodegrade in the body, the lack of toxic reactions and high biocompatibility, this natural polymer has several health-beneficial effects, including strong antimicrobial, antioxidant and anti-inflammatory effects, and as a potential new-generation drug it also has anticancer effects [162]. Chitosan is an excellent candidate for use in the treatment of difficult-to-heal wounds, including chronic wounds; therefore, the use of chitosan-based dressings is one of the most common applications of this polymer in clinical practice [163,164,165]. In addition, it has been shown to contribute to the tissue repair process through the mechanism of increased infiltration of polymorphonuclear neutrophils and macrophages into damaged tissue and the activation of complementary, normal fibroblasts, as well as increased granulation tissue and angiogenesis [166]. Wound healing is a dynamic and complex process where prevention of infection is essential. Many important changes occur during healing, from inflammation, through to cell migration, angiogenesis and cell matrix synthesis, to collagen deposition and re-epithelialization. Literature data supported by numerous studies indicate that chitosan effectively supported cell growth due to its high positive surface charge. Free amino groups could form complexes with acidic groups of blood cells, which supported hemostatic processes, stopped bleeding and consequently accelerated the entire physiological process of wound healing. The coagulation activity of chitosan is variable, depending on its average molar mass and the degree of deacetylation, as these parameters have a significant impact on the final cationic properties of this polymer. Higher molecular weight has been found to promote coagulation [167]. Oxidative stress is the main cause of many diseases, such as cancer, damage to the immune system, cardiovascular diseases, etc. Oxidative reactions can cause free radicals that damage the structure of cells, disrupt their functions and cause damage to the tissues of many vital organs. Chitosan and its derivatives have been shown to be a good natural antioxidant that can inhibit these processes [168,169]. Chitosan has the ability to remove excess free radicals in the body, increase the activity of antioxidant enzymes and inhibit the process of lipid peroxidation. This action supports the natural antioxidant capacity of the body and delays the cellular aging process. Studies have shown that the antioxidant properties of chitosan are closely related to its average molar mass and degree of deacetylation. Literature data show that chitosan with a lower molar mass and/or a high degree of deacetylation shows better antioxidant activity [170]. The antioxidant capacity of chitosan is probably related to the active free groups in its molecule [170,171]. The antioxidant activity of chitosan is strongly related to its molar mass. It was found that it increases with a decrease in its molar mass, while chitosan with a high molar mass does not show antioxidant activity, or the activity is negligible, mainly against superoxide and hydroxyl radicals. This phenomenon is due to the fact that shorter chains are less susceptible to the formation of intramolecular hydroxyl bonds, which results in increased reactivity of hydroxyl and amino groups, which contribute to the “scavenging” of radicals in cells [172]. Therefore, there is a chance to include chitosan in nutritional practice, which may bring health benefits, especially in the case of typical age-related ailments and the natural process of cellular aging [173]. Chitosan also has an immune-stimulating and anti-inflammatory effect. Inflammation is the host’s natural biological and immune response to specific stimuli such as infections, other foreign agents, and physical or psychological stress. It involves a wide and complex set of reactions involving various biological systems mediated by a huge number of endogenous molecules. It has been shown that chitosan and its derivatives are able to interfere in this physiological process at several of its stages. The ability of chitosan and its derivatives to modulate the immune response seems to be related to the presence of N-acetyl-D-glucosamine and the activation of membrane surface receptors and cellular response pathways of the immune system [174,175]. The cells secrete various cytokines that regulate downstream response pathways and play a fundamental role in signaling between cells of the immune system. Studies have shown the ability of chitosan and its derivatives to stimulate the production of cytokines by macrophages, which are involved, among others, in nonspecific host resistance to viral and bacterial infections. The process of formation of cancer cells involves the proliferation of abnormally transformed cells, which divide uncontrollably and gain the potential to invade nearby normal cells or migrate to other tissues and organs. Current cancer treatment is mainly based on surgery and chemotherapy. Although chemotherapeutic drugs can effectively kill cancer cells, the side effects caused by anticancer drugs impose numerous limitations on many patients. At the same time, the immune system is activated, stimulating the destruction of cancer cells [176,177,178]. Free amino groups in chitosan molecules play an important role in the fight against cancer. Studies conducted on three cancer cell lines—human cervical cancer cells (HeLa), human liver cancer cells (Hep3B) and human colon cancer cells (SW480)—show that chitosan with a higher positive charge showed significantly higher anticancer efficacy [179]. Many *in vitro* studies conducted so far have also revealed the inhibitory effect of this biopolymer on a significant reduction in cancer cell growth and tumor mass. An example is the inhibition of the growth of human hepatoma cells (HepG2) by increasing the expression of the caspase-3 protein, which thus induced cell apoptosis [180]. Treatment of human colon cancer cells (HT-29) with a low dose of chitosan resulted in an increase in the production of some antioxidant enzymes, including glutathione, glutathione S-transferase and quinine reductase, indicating the antitransformative and chemopreventive effects of chitosan [181]. The anticancer activity of chitosan involves the influence of several mechanisms in different phases of carcinogenesis. In the initial phase, the polymer inhibits COX-2 expression and NF-kB also inhibits inflammation, while increasing the level of the antioxidant enzyme, and AMPK activity prevents the transformation of normal cells into abnormal cells [182]. As the tumor develops, chitosan reduces tumor mass growth by inhibiting the expression of proteins related to cellular metabolism, including mTOR, catenin B, pyruvate kinase, and ornithine decarboxylase, as well as activation of the caspase-3 and IL-12 pathways, inducing cell apoptosis [180].

#### 3.3.1. Tissue Engineering

The requirements for scaffolds used in tissue engineering include compatibility and compliance with the human body, nontoxicity and biodegradability, as well as the lack of allergenic and inflammatory reactions. During the production of bioscaffolds, attention should also be paid to the appropriate mechanical properties, morphology and porosity, supporting the healing process and tissue renewal. Due to its numerous biological properties, chitosan is a suitable material for the construction of bioscaffolds used for the reconstruction of human organs and tissues. It can be used to produce, for example, hydrogels, microcapsules, membranes, sponges and nanofibers, which can be used for various regenerative purposes [183]. The porous structure obtained from chitosan enables cell multiplication, migration and the exchange of nutrients. This morphology is beneficial in the process of angiogenesis in promoting the survival and function of the regenerated soft tissue. In order to adapt the properties of the scaffold to the requirements of the diseased tissue, chitosan must be subjected to appropriate physical and chemical processes [184]. The low mechanical strength of chitosan biomaterials can be overcome by introducing inorganic materials to the bioscaffolds used, which will strengthen them. Frequently used inorganic compounds, such as calcium carbonate, calcium phosphate and silica, have proven themselves as a composite for chitosan scaffolds. Additions of other polymers may also be a solution. In order to increase the mechanical strength and develop the internal structure of chitosan-based biomaterials, silk, alginate, PLA, HA and bioactive nanomaterials such as hydroxyapatite, SiO_2_, TiO_2_, ZrO_2_, etc., were added [185].

The most important areas of tissue engineering using chitosan for the production of biomaterials are the engineering of cartilage and bone tissue, blood vessel tissue, regeneration of the cornea, skin tissue and periodontal tissue engineering. Bioscaffolds made of chitosan may promote cell adhesion to the surface, their differentiation and proliferation. Chitosan resembles the structure of glycosaminoglycans (GAGs); therefore, it can mimic GAGs in the regulation and modulation of many bioactive factors. Chitosan meets these requirements and can replace missing or damaged tissues and organs, and enables cell adhesion and proliferation [186]. Chitosan belongs to a large group of biopolymers and is still gaining popularity among scientists around the world. Due to its unique features, in recent years it has become a valuable biomaterial for many applications in tissue engineering and regenerative medicine. It is very competitive for the solutions used so far, mainly due to the possibility of obtaining it on a large scale and at low costs, as well as valuable physicochemical and biological properties. Its positively charged and reactive functional groups enable the formation of complexes with anionic polymers, including proteins, which largely affects the ability to regulate cellular activity. Moreover, chitosan is biocompatible, hemocompatible and nonimmunogenic. In the body, it is degraded to nontoxic oligosaccharides by the action of lysozymes and has strong antibacterial properties [187]. Numerous studies on chitosan biomaterials have confirmed the lack of causing inflammatory and allergic reactions in the human body after implantation, injection, topical application or ingestion. Chitosan has the properties of stimulating the wound healing process and the regeneration of soft and hard tissues. Chitosan scaffolds are used in the regeneration of skin, liver, bone and cartilage tissues, heart tissue, corneal tissue and vascular tissue [188].

#### 3.3.2. Antibacterial Activity

Chitosan is known for its antimicrobial activity. Due to its polycationic chemical nature, it is a promising candidate as a natural antimicrobial agent. The positively charged amine group electrostatically interacts with the negatively charged membranes of microorganisms, leading to leakage of cellular contents and death of the bacterial cell. Lipopolysaccharides and surface proteins present on the bacterial cell surface interact with the active groups in the chitosan chain [189]. Molecular weight, degree of deacetylation, chitosan concentration and pH of the solution significantly affect the antibacterial activity of chitosan. Chitosan with a high molecular weight and a high degree of acetylation works more actively. The hydrophilicity and the content of negatively charged molecules present in the microbial cell membrane also affect the interaction with chitosan, which has a stronger effect on Gram-negative bacteria compared to Gram-positive bacteria. The mechanism of this bactericidal effect is also associated with the inactivation of bacterial enzymes and the substitution of metal ions, as well as the interaction with teichoic acid on the bacterial cell surface. Chitosan with a high molecular weight is able to form a polymer film with the microbial cell wall, which, among other things, prevents the delivery of nutrients and the death of bacterial cells. Low-molecular-weight chitosan has the ability to penetrate inside cells, where it combines with negatively charged intracellular components, e.g., phosphate residues of DNA molecules, which blocks reactions at the level of transcription and mRNA synthesis. The antibacterial effect is related to the structure of the microbial cell wall. High-molecular-weight chitosan forms an envelope around the thicker cell walls of Gram-positive bacteria. Gram-negative bacteria have a much thinner cell wall. Chitosan with a low molecular weight works on them most effectively, penetrating inside and disintegrating their genetic material [190,191]. Numerous literature data have shown the effective antifungal activity of chitosan against *Candida albicans* and *Fusarium solani*. The fungicidal mechanism of chitosan is the active penetration of chitosan molecules into the hyphae of the fungus, where the structure of the enzyme necessary for the growth of the fungus is destroyed [192].

Commonly used indicators for assessing the antibacterial activity of chitosan preparations are MIC, MBC and MFC. MIC (minimum inhibitory concentration) is the lowest concentration of a biocide that inhibits the growth of microorganisms, expressed in mg/L. The parameter determines what concentration of the active substance inhibits the growth of bacteria or fungi. MBC (minimum bactericidal concentration) is the lowest concentration of the bactericide at which 99.9% of bacteria are killed, expressed in mg/L. The parameter informs about the concentration of the preparation that has bactericidal activity and directly destroys vegetative forms. MFC (minimum fungicidal concentration) is the lowest concentration of fungicide at which 99.9% of fungi are killed, expressed in mg/L. All these parameters are determined for a given genus and species of bacteria and fungi, used in in vitro studies. Chitosan has antifungal activity against, among others, *Aspergillus niger*, *Candida albicans*, *Candida glabrata* and *Fusarium solani* (fungi); and antibacterial activity against, among others, *Staphylococcus aureus*, *Escherichia coli*, *Salmonella typhimurium* and *Pseudomonas aeruginosa* [193].

#### 3.3.3. Chitosan Hemostatic Dressings

Hemocompatibility is defined as the interaction of the foreign material with the blood. A biomaterial is hemocompatible when it exhibits less than 5% hemolysis. Literature data report good hemocompatible properties of chitosan. Chitosan-based materials have been found to be highly hemocompatible and not hepatotoxic. In addition, chitosan-based biomaterials induce blood coagulation processes and exhibit significant hemostatic properties. In the last few years, the hemostatic effect of chitosan in relation to physiological wound-healing processes has been extensively studied. The material intended for this purpose should be nonthrombogenic towards plasma proteins and blood cells that can activate a thrombotic or hemolytic response. Chitosan can activate and replenish the blood coagulation system [194]. Chemical modification strategies can be used to improve the blood compatibility of chitosan. N-succinylchitosan (N-SC) has been found to have very high hemocompatibility, in addition to having anticancer effects, and may support the immune system for cord blood transplants [195]. The hemostatic properties of chitosan are significantly influenced by the molecular weight and degree of deacetylation. A higher degree of deacetylation improves erythrocyte and platelet aggregation, which is important in initiating hemostasis [196]. The cationic nature of chitosan allows it to induce hemostasis. Platelet and erythrocyte surfaces are negatively charged due to the presence of phosphatidylcholine, phosphatidylethanolamine and sialic acid groups. The presence of negative charges on platelets and erythrocytes causes electrostatic repulsion and hinders the aggregation process. Amine groups present in chitosan facilitate the aggregation of erythrocytes, interacting electrostatically with its surface charges. Inducible hemostasis occurs after platelet activation [197]. Currently, there are many chitosan-based hemostatic dressings available on the market, including Chitogauze, Celox Gauze, Mini-sponge dressing, Hemcon, Trauma Gauze and ChitoFlex. Due to the high biocompatibility and additionally effective antibacterial effect, chitosan dressings are more and more often used in the treatment of hard-to-heal wounds and burns. Chitosan-based hemostatic dressings have been approved by the FDA [198].

### 3.4. Application of Chitosan in Agriculture

The widespread use of agriculture is related to the property of chitosan to act as a trigger in plant defense against pathogenic microbes, exhibiting a broad spectrum of antimicrobial activity against bacteria, fungi and viruses. However, chitosan is more effective against fungi than bacteria, and often has a stronger inhibitory effect against Gram-positive bacteria than against Gram-negative bacteria [199]. It can effectively inhibit the development of phytopathogenic fungi at different life-cycle stages [200]. The effectiveness of chitosan depends on its parameters such as Mw, DD, concentration and combination with other ingredients. Although increased efficacy observed against fungi such as *B. cinerea* or *F. oxysporum f.* sp. *radicis-lycopersici* [201]. Chitosan with the same concentration but different Mw (92.1 kDa and 357.3 kDa) can act with different mechanism. In tests against Penicillium italicum, lower Mw showed effectiveness in inhibiting fungal growth, while higher Mw showed stronger antifungal activity [202]. The Mw of chitosan affected the differential when applied alone and in combination; for example, with oligosaccharides, synergistic effects against a seed-borne pathogen such as *F. graminearum* were observed [203]. An antifungal effect was activated by applied chitosan with metal particles as a sole ingredient [204]. Chitosan nanocomposites with silver nanoparticles were tested as an antifungal agent against Rhizoctonia solani, Aspergillus flavus and Alternaria alternata isolated from chickpea seeds [205]. Chitosan also affects germination and hyphal morphology fungal of pathogens threatening harvested crops (e.g., *Botrytis cinereal*, *Rhizopus stolonifer*) [206]. A different use of chitosan in agriculture is applied. Chitosan treatments as a solution for seed soaking or coating provide an antifungal seed treatment, providing protection against seed-borne or soil-borne pathogenic fungi, which can significantly lower seed germination and plant emergence in the field [207,208]. Furthermore, chitosan as a polymer for protection was used as a film on postharvest decay of fresh fruits and vegetables [209,210].

### 3.5. Other Important Applications

#### 3.5.1. Water and Wastewater Treatments

Another wide application of chitin is industrial water and wastewater treatment. All over the world, research is being carried out to find the best methods and the most effective absorbers for removing harmful contaminates such as dyes, metal ions, i.e., Cu^2+^, Ni^2+^, Zn^2+^, Cd^2+^ and Pb^2+^ from water. The range of absorbers is very wide. They can be divided according to their synthetic or natural origin. The main criterion for selecting the material for the application, however, is always the drastic reduction of the concentration of the pollutant and the economic aspect of the process. Among the natural absorbents, chitin is also used, and chitosan is used alone or in the chitin–lignin system [211,212,213,214] or in composites, e.g., with green algae powder (*Ulva lactuca*) [215]. The last one is very effective for bioremediation of Cd^2+^ ions. The methods used to remove or neutralize pollutants include adsorption methods as well as coagulation/flocculation technology. The adsorption method is extremely flexible and easy to design and implement. It is characterized by high process efficiency at low cost, and thanks to its use, it is possible to remove both organic and inorganic impurities. Coagulation/flocculation technology is also another low-cost method used to remove or neutralize pollutants, and at the same time is often the only one possible to use in a given industry. This process consists of two stages: the addition of coagulants leads to physical and chemical reactions that lead to the destabilization of colloidal particles. In the second stage (flocculation), as a result of collisions and transport of destabilized particles, the so-called “flocculation” can be removed from the water by filtration or sedimentation/flotation. A flocculant based on chitosan with mercaptoacetic acid is known, tested for the removal of metal ions from wastewater and the reduction in system turbidity [216].

#### 3.5.2. Textile Industry

Another wide application of chitosan is to use it for coating, e.g., on textile substrates. Chitosan, as a cationic polymer, has an ion affinity to textile fibers, which are mostly anionic. [217]. Textiles are much more widely used beyond their traditional use for dressing people and homes. It is possible to introduce chitosan into textiles to provide special properties, such as additional protection in adverse environments, specific resistances, comfort, and aesthetics. New attributes change performance and consumer demands and increase industry competitiveness. There are many techniques that result in the transformation of ordinary fabric into a fabric with the desired performance properties. Huge advances in polymer science are supporting the growth of the use of coatings. An effective and commercially applicable technology for the production of functional textile coatings is microencapsulation. It is mainly used for drug delivery [218] as well as for applications in civil construction, hospitals and as a carrier for cosmetics. The versatility of the microencapsulation technique allows it to be used with perfumes, cosmetics and personal care products [219,220], food [221,222], pharmaceutical and medicinal treatments [223,224], pest and insect repellents [225], environmental recovery [226], construction [227] and textile [228].

#### 3.5.3. Pulp and Paper Industry

The pulp and paper industry are a very important sector contributing to social and economic advancement. Therefore, there is a need and great opportunity to conduct extensive research in order to produce cheap paper and pulp products and additives with higher and improved properties, i.e., grease resistance, wet and dry strength, etc. [229]. Chitosan and its derivatives appear to be good retention and drainage agents, which are important wet-end additives in the papermaking process. Research focuses on the use of chitosan as a papermaking additive to improving the wet and dry strength of paper and as a color-fixing agent for colored papers [230]. The inherent antimicrobial and film-forming properties of chitosan allow for the fabrication of functional antibacterial and greaseproof papers [231]. Wang et al. described a method for the preparation of chitosan/montmorillonite coatings for the production of oil-resistant papers that can be safety used in the food industry [232]. Promising materials for food packaging were successfully fabricated papers with enhanced barrier and antibacterial properties by sequential deposition multilayers consisting of chitosan and carboxymethyl cellulose on a paper surface. The multilayers effectively improved air, water and oil resistance, and also showed antibacterial activities against *E. coli* and *S. aureus*. [233]. Chitosan is also used as a chelating and complexing agent in the treatment of pulp and paper wastewater for removing lignin, undesirable impurities and dyes, and to lower total organic and chemical carbon oxygen demand [234].

## 4. The Life Cycle Assessment of Chitin and Chitosan in Relation to Circular Economy

From year to year, more and more products are produced from petrochemical polymers. It has been estimated that the residents of Warsaw use about 1.8 million disposable plastic bags in one day [235]. All countries in the world individually control their own ecological policy, which defines the principles of waste management in accordance with the idea of sustainable development. In addition, in European countries, the European Commission announced in 2015 that over the next few years it will take a number of measures to mobilize Member States to transition to a circular economy, where the generation of waste will be limited to a minimum. The circular economy, i.e., a resource-saving, low-emission and sustainable economy, aims not only to increase people’s awareness of improving the level of environmental protection, but above all to strengthen the competitiveness of enterprises in EU countries. The action plan prepared by the European Commission identifies four main areas for the transition to a circular economy. These are production (including the design of products as well as their manufacturing processes), consumption, waste management and stimulating the market for secondary raw materials and water reuse. The European Commission has also identified five priority areas: plastics; food waste; critical raw materials; demolition and construction waste; and biomass and biobased products [236,237]. In 2020, 4.808 tonnes of waste per capita was generated in the EU countries (the most waste was generated in Finland—20.993 tonnes per capita, and the least in Croatia—1.483 tonnes per capita). Poland ranked 13th (just behind France and Germany) with 4.492 tonnes of waste per capita. In the same year, in the EU countries, 31.3% of waste was landfilled and 39.2% was recycled [238].

The priority areas in the circular economy identified by the European Commission, the biomass and bioproducts sectors, are very important. The chitosan market is one of the strictly confidential markets. According to available data, the global chitosan market in 2019 was valued at USD 6.8 billion. In line with the trend observed over the years, it was estimated that the value of this market in 2020–2027 will increase with a cumulative annual growth rate (CAGR) based on revenues of 24.7%. Below in Figure 4 is presented the estimated demand (in tonnes) for in the United States in 2016–2027 [239].

The growing demand for various chitosan-based products used in water treatment techniques, medicine, veterinary medicine, pharmacy, cosmetics, beverages and food indicates that these industries will drive the growth in demand for this polymer, and its annual production from chitin alone and crab and shrimp alone is 2000 tonnes [240]. The main sources of obtaining chitin are presented in Figure 5. The greatest amount of chitin in the form of dry matter can be obtained from krill and crabs [241].

LCA (life cycle assessment) is a standardized practice of assessing the effects of a product on the environment during its entire life cycle (from the collection of the basic material/raw material to the final use, i.e., the so-called “grave to cradle analysis”), by measuring the growth efficient use of resources as well as reducing environmental burdens (liabilities). A product, e.g., a biopolymer, has several stages of life, and each of them has quite a significant impact on the ecosphere. The LCA consists of the following elements: quantification and identification of environmental burdens, e.g., materials and energy used as well as emissions and wastes released into the environment, assessment of the potential impacts of these burdens and estimation of available options to reduce them [242].

The general life cycle of a product includes stages that start with the acquisition of the raw material, which is processed, followed by product manufacturing, packaging, transport, use, maintenance, disposal and recycling (Figure 6). Each of the listed stages of the product’s LCA can generate potentially dangerous, single-use substances, toxicological stress for human health, large water resources, air, etc. Product life cycle management is helpful in maintaining ecological balance by adjusting the consumption of raw materials and energy (e.g., fuels) and reducing environmental risks [243]. The LCA is a defined unit to maintain organic quality and integrity to determine the optimal product and processes in terms of their new growths [240]. In the area of chitosan life cycle assessment (LCA), a peer-reviewed study was identified that concerned the process related to this biopolymer.

Several ways of collecting microalgae by flocculation have been evaluated by using, e.g., chitosan (produced from crustacean shells), ferric sulfate and alum. Based on the above studies, chitosan has been proven to be much better from the point of view of taking care of the environment compared to ferric sulfate and alum as a flocculant used to harvest microalgae [240,243,244]. The life cycle study was based on original information obtained from two real manufacturing companies located on two continents: Europe and Asia (Mahtani Chitosan company, off the coast of Gujarat). The aim of this study was primarily to consider two “looks” at chitosan supply chains and also to understand their so-called critical points—hot spots. The LCA study of the above chitosan was carried out in accordance with EN ISO 14040 (2009): Environmental management—Life cycle assessment—Principles and structure and EN ISO 14044 (2009): Environmental management—Life cycle assessment—Requirements and guidelines [240,245]. One of the partners that largely funded the chitosan LCA studies was the European Commission. In addition, both chitosan producers involved in the project were interested in the research results, primarily to define strategies for improving the environmental profile of their product range. Asian and European biopolymers were not assessed in terms of ecological properties, because they were ultimately directed to different markets (Indian polymer was intended for agricultural applications, produced in the amount of 50 tons per year), while Europe was focused on the production of chitosan for use in the medical sector (due to confidentiality, the company was obliged to remain anonymous) [240]. The raw material for the polymer from India were shrimp shells from fisheries in the Arabian Sea. Shrimp waste was transferred from seafood processing plants to a company producing polymers and then processed into the starting polymer—chitin. In order to obtain it, transported waste from the food sector was subjected to a demineralization process using diluted HCl and deproteinization using a diluted NaOH solution. After the protein removal processes, the obtained chitin was subjected to a deacetylation process using very concentrated alkalis (40–50% NaOH solutions) under “hot” conditions. The processes described above produced wastewater, which before being released into the sea was treated in on-site treatment plants. The sludge, or extracted protein, in the initial biopolymer purification process was recycled and partly used as fertilizer for crops (a potential source of nitrogen), while the calcium salts were either dumped in landfills or used as building material or for repairing damage to roads in areas included in composition of the company [240].

The life cycle of the European producer of chitosan began in Canada (Newfoundland), where the snow crab (Chionoecetes opilio) was obtained, and more precisely its shell, which was dried and transported to China (to Qingdao). There, chitin was produced from them (as a result of the process of demineralization and deproteinization, which have already been described in the Indian supply chain). Then, the obtained chitin was sent to a European chitosan producer, where it was subjected to the deacetylation process. The production of chitosan has had an impact on the animal feed market, where protein is produced as a waste in the form of sludge, the production of which results in the creation of an equivalent amount of energy and feed protein. According to the data contained in Feedipedia and on the basis of the documentation of the Mahtani company, it was estimated that from 1 kg of snow crab shells (moisture content 75%) can be obtained the equivalent of 2.1 MJ of energy obtained from feed and 0.16 kg of protein equivalent [240,246,247]. In order to obtain 1 kg of pure chitin, it was necessary to process 33 kg of wet chitin shells. Shell transport was carried out using specialized trucks with open trailers, which used 1.4 l of oil per 1 ton of shrimp waste; in the processing process at the production plant, the waste bulldozer used 0.02 l of diesel fuel per 1 kg of chitin, and moreover, 8 kg of 32% hydrochloric acid solution, 1.3 kg of soda lye, and 1.3 kWh were used per 1 kg of pure polymer electricity and 167 L of fresh water [240,247]. A total of 0.7 kg of CO_2_ per kg of chitin was released from the shell calcium carbonate during acid treatment; the solid waste from the production contained 1.5 kg of calcium salts, which were landfilled, and the dry weight of the protein sludge used as a component of fertilizers was 4 kg. The use of such a fertilizer substitute reduced the consumption of synthetic mineral fertilizers, where 1 kg of nitrogen in the protein by-product in chitosan production replaced 0.4 kg of nitrogen contained in mineral fertilizers [248]. According to the company from India, 1 kg of chitosan was obtained from 1.4 kg of purified chitin. To produce 1 kg of Indian chitosan, 1.4 kg of chitin was required, and the following auxiliary agents were used: 5.18 kg of NaOH, 31 MJ of wood fuel, 250 l of water and 1.06 kWh [240,246,247]. Wastewater, which was formed at the stages of chitin and then chitosan production, was first neutralized and then subjected to the processes of initial settling on sieves and biological treatment with the use of activated sludge, and then subjected to sand filtration. After wastewater treatment, it was then discharged into the sea [239]. Methods of composting snow crab waste have been used to produce compost, which is then used as fertilizer. Based on the data provided by GAMS [240,249], the estimated transport distance of snow crab shell waste to the heaps at the composting facility was approximately 25 km. Gas emissions in the composting process, i.e., carbon dioxide, nitrous oxide, methane, ammonia, nitrogen oxides and hydrogen sulfide, were estimated using mass balances based on the composition of snow crab waste reported by GAMS and other literature sources [240,250]. It was found that 1 kg of phosphorus obtained from composting crab waste replaced 0.95 kg of P in fertilizer of synthetic origin [248]. Based on the statistics of the European chitosan producer obtained from the Chinese chitin supplier, it was determined that the production of 1 kg of pure chitin required 10 kg of dried snow crab shell, 1.2 kWh of electricity, 6 kg of coal for heating, 9 kg of 6% HCl solution, 8 kg of 4% NaOH solution and 300 l of fresh water. Gas emissions, e.g., CO_2_ as a result of acid treatment of shells, was estimated at 0.9 kg of CO_2_ per 1 kg of polymer [240]. Figure 7 presents a system of products for obtaining general-purpose chitosan in Asia and Europe.

## 5. Conclusions

The review of chitosan modification methods presented in the manuscript and their great variety prove that currently, a number of new compounds with significantly different chemical and physical properties can be obtained from one biopolymer. The main applications of chitin and chitosan use their positive functional properties, obtained through the possibility of modifications. Usable properties include better physicochemical and biological effects, i.e., the ability to dissolve in water and nontoxic solvents, antibacterial properties, biocompatibility or nontoxicity. Additionally, as a natural polymer, it is fully biodegradable, which does not have a negative impact on the environment. The molecular structure of chitosan, thanks to active functional groups, can be modified to form derivatives with improved physical, chemical and/or physiological functions. Chitosan derivatives are created by various chemical reactions and are widely used in many areas of our lives. Chitosan as a natural polymer is one of the main representatives that are used to design various advanced structural and functional materials using the principles of bionics found in the natural environment. Due to the ability to biodegrade in the body, the lack of toxic reactions and high biocompatibility, this natural polymer has several health-beneficial effects, including strong antioxidant and antimicrobial, antioxidant and anti-inflammatory effects, and as a potential new-generation drug, it also has anticancer effects and it is perfect for transplantation medicine as scaffolds, prostheses, etc. Chitin, chitosan and their derivatives are among the active ingredients also widely used in the cosmetics industry (preparations for skin, hair, gums and teeth care). Thanks to its bioactivity and anti-inflammatory effects, accelerating healing, chitosan has found application in dentistry. Chitosan is used in the food industry, playing the role of a food preservative, acting as a natural antioxidant and improving the quality of food safety and shelf life. Its common use in agriculture is related to the property of chitosan as a factor that triggers the body’s defense against pathogenic microorganisms, showing a wide spectrum of antimicrobial activity against bacteria, fungi and viruses. The study of the life cycle of chitosan proved that it is a safe biomaterial from the point of view of taking care of the environment. The growing demand for various products based on chitosan indicates an increase in demand for this polymer, among many industries. Therefore, its increasingly common use and successive work of researchers on the improvement of many functional properties may in the future completely replace many artificial polymers used so far. Biopolymers have emerged as an effective solution to replace petroleum-based polymeric materials and reduce dependence on petroleum reserves. Chitosan is one of the polymers of the future that keeps researchers interested. For many years, new works have been published that describe new ways of obtaining various forms of the polymer after applying modifications, thanks to which chitosan can find more and more new applications.

## Figures and Tables

**Figure 1 polymers-15-00793-f001:**
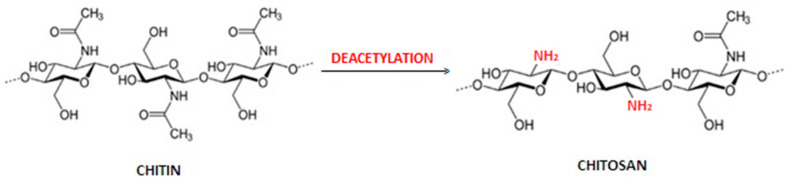
The chitin deacetylation reaction. The deacetylation process and the reactive amino groups formed in it are marked in red. Authors’ own figure from Reference [4].

**Figure 2 polymers-15-00793-f002:**
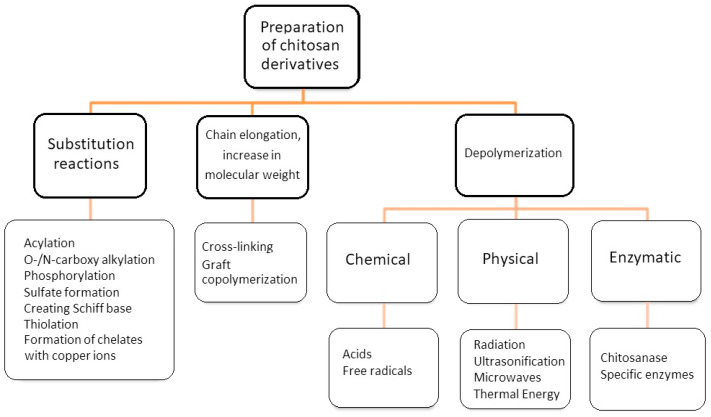
Scheme of reactions leading to the production of chitosan derivatives. Authors’ own figure from Reference [36].

**Figure 3 polymers-15-00793-f003:**
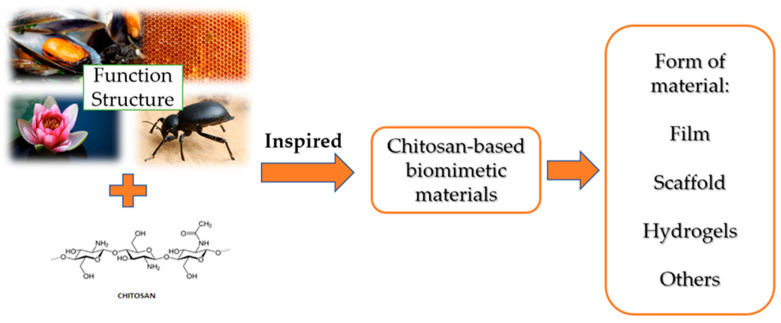
Scheme of chitosan-based biomimetic materials and the forms of material prepared. Authors’ own figure from Reference [75].

**Figure 4 polymers-15-00793-f004:**
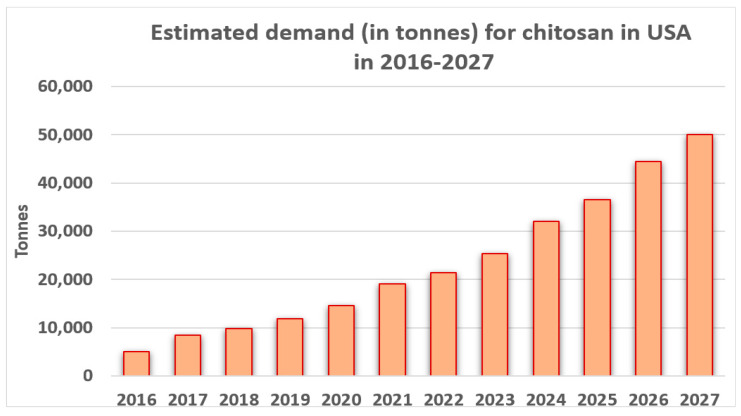
Estimated demand (in tonnes) for chitosan in the United States in 2016–2027. Authors’ own figure from Reference [239].

**Figure 5 polymers-15-00793-f005:**
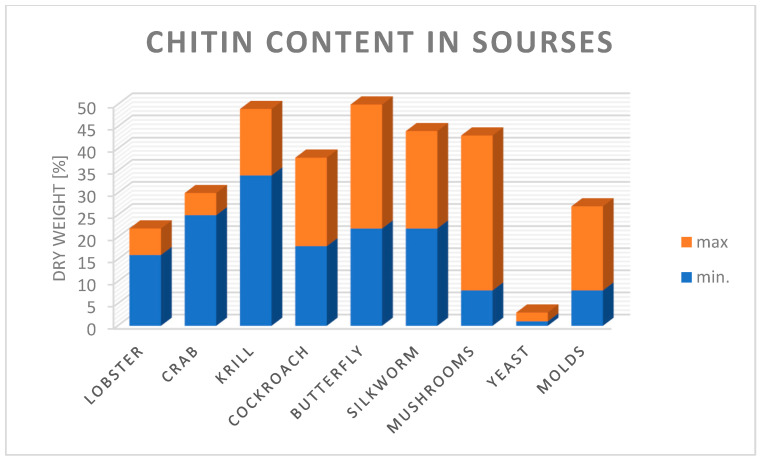
Chitin content in various organisms, in the form of dry matter from minimum to maximum % dry weight chitin content. Authors’ own figure from Reference [241].

**Figure 6 polymers-15-00793-f006:**
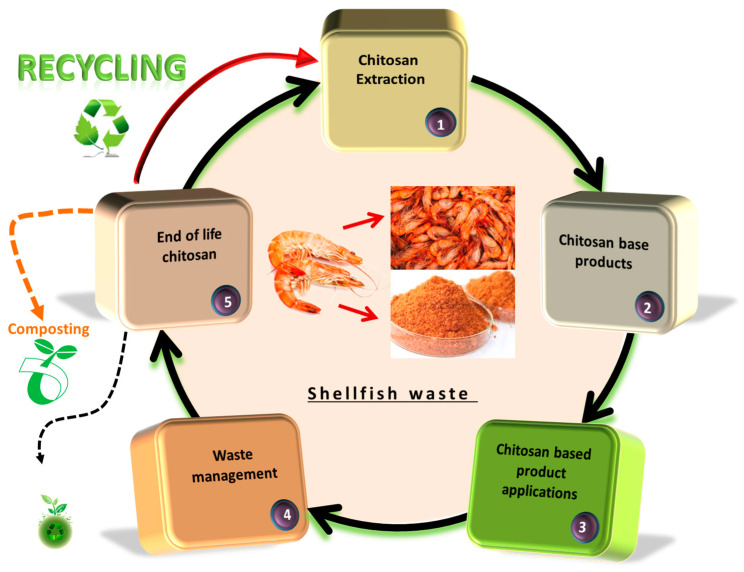
The general scheme of life cycle assessment of chitosan.

**Figure 7 polymers-15-00793-f007:**
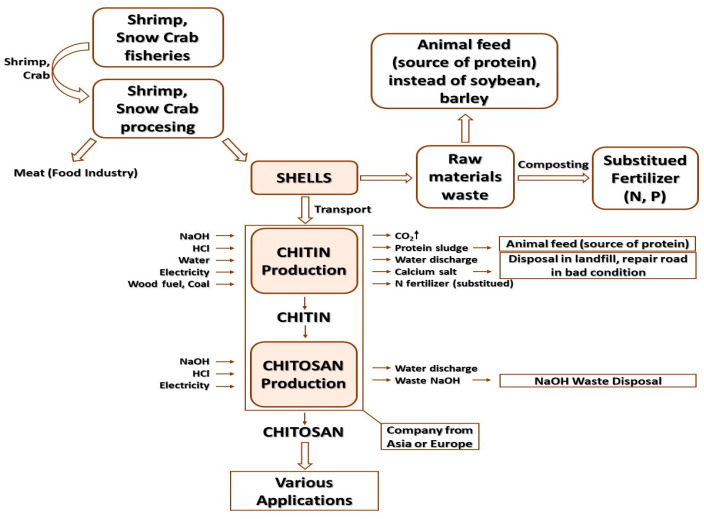
System of products for obtaining general-purpose chitosan in Asia and Europe. Authors’ own figure from Reference [240].

**Table 1 polymers-15-00793-t001:** Examples of chitosan blends applied in food packaging.

Additive		Form	Result	Ref.
Polysaccharide	Starch	Films	Improvement in water barrier properties along with enhanced antioxidant and antimicrobial activity	[145,147]
	Cellulose	Films	Improvement in mechanical properties due to electrostatic interactions between the polymers	[148,149]
	Alginate	Films	Very good gas exchange and water vapor transmission	[150]
	Pectin	Films	Helps to maintain physicochemical and sensory values	[151]
Synthetic polymers	Polyvinyl alcohol (PVA)	Films	Improvement in mechanical properties and enhanced barrier performances towards water and oxygen	[152]
	PLA/starch	Films	Improvement in flexibility and thermal properties	[153]
	Low-density polyethylene	Films	Improvement in moisture barrier properties	[154]
Other	Silver	Films	Improved antibacterial properties	[155,156]
	Zinc oxide	Films	Improvement in moisture barrier, mechanical strength and antimicrobial activity	[157]
	Extracts from bee secretions (beeswax and propolis)	Films	Maintains food quality, both visual appearance and taste	[158]
	p-Coumaric acid (p-CA)	Not given	Partially enhanced water solubility and antioxidant property	[159]

## Data Availability

The data presented in this study are available on request from the corresponding author.

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
