# Peer review of "Chitin and Chitosan as Polymers of the Future—Obtaining, Modification, Life Cycle Assessment and Main Directions of Application"

_polymers, 2023, doi:10.3390/polym15040793_

Round 1

Reviewer 1 Report

This review has no novelty; previously, many other authors wrote on this topic. There is no special attraction point for readers. All information’s in general. The authors inserted references in the abstract. The abstract should be unique. Very strange for me in this paper. It is not standard to publish in a polymers journal. As per the title, there is no point in the review. All section has in general information. the conclusion has no reliable information. There is no update of recent references (2022). Authors need to write as per the title. No future direction for this review. No table in the manuscript. Overall, this review is not worth publishing in a polymers journal.

Author Response

Dear Reviewer,

Thank you for the review and valuable comments contained in it, we tried to changed all linguistic shortcomings and add accurate formulations, not only in conclusions. The abstract has been changed. There was a need to refine the manuscript. We have extracted a new section: 2. Biomimetic materials as inspiration for functional materials with chitosan for a general description by readers of this role of chitosan. That’s way the chapter numbering has been changed and the section 2.2 is now 3.2 Application in food packaging and has been expanded and enriched with a description of studies from (approx. 20) nowelty recent literature reports, and the table was added. Manuscrypt in section 2.3 Application of chitosan in medicine” - which is currently no. 3.2. was enriched and divided with additional subsections: 3.3.1 Tissue engineering, 3.3.2 Antibacterial activity, 3.3.3 Chitosan hemostatic dressings.

Section 2.5 is now “3.5 Other important applications” and was enriched and divided with additional subsections: 3.5.1. Water and wastewater treatments, 3.5.2 Textile industry, 3.5.3 Pulp and paper industry, where more newest information about currently research were included.

Extensive use of chitosan and many years of research on its application form and modification, thanks to which we gain great importance in the world of science and are able to provide our review work. Life cycle assessment of chitin and chitosan in relation to the circular economy is a novelty in this type of review articles. We added figures summarizing the manuscript and its widely shown application forms, and statistical data study graphs, because despite this critical assessment, there is a place for our article in the journal Polymers, and above all, we confirmed the statement of the study. Polymer is constantly gaining interest, as evidenced by the number of shares in literature from the last 2 years.

We request you to re-read the manuscript again. 

Reviewer 2 Report

In this review, authors have reported importance of chitin/chitonan polymer on different fields. In my opinion,  the manuscript is well-detailed, and possible mechanisms of polymer modifications are included in the manuscript. However, I would suggest to add several points to improve the text.

- Although authors have mentioned about health applications, in my opinion, some points are missing out. For example, one of the main chitosan research interest in literature is bone tissue engineering. Several recent studies have shown that chitosan has a good impact on osteoblasts. Another point would be highlighting its antibacterial properties in a detailed way.

-Chitosan is also important polymer for the biomimetic studies (e.g.  molecular structure, mechanical properties or surface patterning) . Biomimetic studies are not mentioned in the current manuscript. I think it would be better to add such a section to show its importance.

Author Response

Dear Reviewer,
thank you for the positive review and accurate comments contained therein, there is indeed a need
to make the manuscript more detailed, have separated the chapter:
Biomimetic materials as inspiration for functional materials with chitosan and supplemented
chapter 2.3 (currently 3.3) with additional subsections: Tissue engineering, Antibacterial activity,
Chitosan hemostatic dressings, where the information mentioned is included.

Reviewer 3 Report

This manuscript focus on the extraction, modification and application of chitin and chitosan, which is of interesting to broad readers. After carefully reading, it was found that this article needs major revisions because several issues and explanations are still need to be clarified.

1.      Please replace “LCA” in the title with its full name.

2.      “chitosan” is suggested to be added as a keyword.

3.      Please revise “When α is in the range of 0.5÷0.8” in line 132 to 134.

4.      Please revise “This The definition of cosmetic describes it as any substrate” in line 281. Please polish the manuscript again to resolve such issues.

5.      More typical figures from references are suggested to be added in the manuscript to show the broad application and performance of chitosan.

6.      Chitosan is a good precursor in design functional packaging for foods. More typical references are suggested to be cited for “2.2. Application of chitosan in food industry”, e.g. Journal of Bioresources and Bioproducts 2022, 7 (2), 85-98; Journal of Bioresources and Bioproducts 2022, 7 (1), 1-13.

7.      “2.3. Application of chitosan in medicine” is suggested to be divided into a few paragraphs.

8.      Chitin is widely applied in many fields. More references are suggested to be cited in “2.5. Other important applications” for broad readers, e.g. Journal of Bioresources and Bioproducts 2021, 6 (3), 223-242; Journal of Bioresources and Bioproducts 2021, 6 (1), 11-25.

9.      Some insightful perspectives are suggested to be added into conclusion section.

10.   Please pay attention to the writing of chemical formula in the references, e.g. ref. 82 and ref.96.

Author Response

Dear Reviewer,

thank you for your positive opinion and accurate comments contained therein.
We have responded to all comments and included them in the appendix.

Round 2

Reviewer 1 Report

Authors improved the manuscript in the satisfactory level. However other major issues need to be solved before publications. My major comments as follows:

In title, first letter of each word should be capitalized.

Line 11, leds?

More keywords need to be provided as per the novelty of this review.

In figure 1 elimination of deacetylation group in the arrow mark. And highlight in the colour.

Lines 118-140, need to be written in the better way as it is important of this review.

Why we need modification? Which modifications of chitosan show better properties and its specific application need to be discuss in the manuscript extensively.

Indicate the recent references in the manuscript; https://doi.org/10.3390/polym15010132 ; Food packaging and Shelf Life 33, 2022, 100904 and International Journal of Biological macromolecules 136 (2019) 661-667.

Section 3.3.2 application of antibacterial activity need to be rewritten by considering the following points. Evaluation of antibacterial method and its mechanism.

English have problem in the manuscript. Check carefully.

Author Response

Dear Reviewer, 

thank you for all your valuable suggestions, as required, individual sections have been changed and bugs have been corrected.

The responses to the comments are attached below.

Best regards, 

Reviewer 3 Report

The manuscript has been revised according to the comments and suggest to be accepted.

Author Response

Dear Reviewer, 

Thank you for your positive review and we are glad that our manuscript was subjected to such apt comments.

Best regards, 

Autors

Round 3

Reviewer 1 Report

The authors modified the manuscript to a satisfactory level.